# EEG Signal Prediction for Motor Imagery Classification in Brain–Computer Interfaces

**DOI:** 10.3390/s25072259

**Published:** 2025-04-03

**Authors:** Óscar Wladimir Gómez-Morales, Diego Fabian Collazos-Huertas, Andrés Marino Álvarez-Meza, Cesar German Castellanos-Dominguez

**Affiliations:** 1TECED—Research Group, Faculty of Systems and Telecommunications, Universidad Estatal Península de Santa Elena, Avda. La Libertad, La Libertad, Santa Elena 7047, Ecuador; 2Signal Processing and Recognition Group, Universidad Nacional de Colombia, Manizales 170003, Colombia; dfcollazosh@unal.edu.co (D.F.C.-H.); amalvarezme@unal.edu.co (A.M.Á.-M.); cgcastellanosd@unal.edu.co (C.G.C.-D.)

**Keywords:** brain–computer interface (BCI), signal prediction, motor imagery (MI), electroencephalography (EEG), regularization analysis, multiple regression analysis

## Abstract

Brain–computer interfaces (BCIs) based on motor imagery (MI) generally require EEG signals recorded from a large number of electrodes distributed across the cranial surface to achieve accurate MI classification. Not only does this entail long preparation times and high costs, but it also carries the risk of losing valuable information when an electrode is damaged, further limiting its practical applicability. In this study, a signal prediction-based method is proposed to achieve high accuracy in MI classification using EEG signals recorded from only a small number of electrodes. The signal prediction model was constructed using the elastic net regression technique, allowing for the estimation of EEG signals from 22 complete channels based on just 8 centrally located channels. The predicted EEG signals from the complete channels were used for feature extraction and MI classification. The results obtained indicate a notable efficacy of the proposed prediction method, showing an average performance of 78.16% in classification accuracy. The proposed method demonstrated superior performance compared to the traditional approach that used few-channel EEG and also achieved better results than the traditional method based on full-channel EEG. Although accuracy varies among subjects, from 62.30% to an impressive 95.24%, these data indicate the capability of the method to provide accurate estimates from a reduced set of electrodes. This performance highlights its potential to be implemented in practical MI-based BCI applications, thereby mitigating the time and cost constraints associated with systems that require a high density of electrodes.

## 1. Introduction

Motor imagery (MI) is a cognitive process that involves the mental simulation of movements without their physical execution, sharing mechanisms with motor execution [1]. Motor imagery, considered an advanced function of the cortex, is emerging as an effective and novel learning tool that offers simulated solutions for motor tasks during the training period [2,3]. In contrast, motor execution (ME) involves the effective practice of movement. MI and ME share sensorimotor areas, and both processes require the planning and execution of an identical motor plan, although they differ in some of their neuronal mechanisms [4]. Commonly, to verify that motor learning has occurred and that it has been maintained after a training stage, the performance of motor execution is evaluated [5,6]. Moreover, the brain plasticity resulting from motor learning is often reflected in significant alterations of the electroencephalographic (EEG) signals [7], particularly in the phase prior to execution [8,9]. However, the ability to acquire new motor skills varies significantly among individuals, mainly due to differences in the structure and functionality of the brain observed in EEG recordings [10,11]. Previous studies have shown that approximately 15% to 30% of users experience difficulties in controlling motor imagery (MI)-based brain–computer interfaces (BCIs), a phenomenon known as BCI illiteracy, which significantly restricts the widespread application of MI-BCIs [12,13]. To address this challenge, techniques such as elastic net and Common Spatial Pattern (CSP) have proven effective in selecting relevant EEG features and improving classification accuracy [14].

The development of brain–computer interfaces (BCIs) based on motor imagery has driven significant advancements in assistive technologies, neurological rehabilitation, and enhancement of motor skills [15]. These systems enable the translation of neuronal activity, recorded through electroencephalography, into operational commands to control external devices, thus facilitating a direct interaction between the brain and its environment [16]. However, the effectiveness of BCIs in practical applications faces major challenges, mainly due to the high interindividual variability in control capability and the quality of EEG signals, which limits their generalization and applicability [17,18].

The variability in BCI control can largely be attributed to differences in brain structure and function among users [19,20]. These differences affect the quality of the captured EEG signals, complicating the motor learning process and the users’ ability to effectively manipulate these systems [21,22]. Moreover, the noisy and non-stationary nature of EEG signals introduces additional complexities in designing robust and reliable BCIs [23]. The high correlation between channels and environmental noise complicates the identification of relevant neural patterns [24]. On the other hand, the quality of the recorded EEG signals is influenced by multiple factors, including electrode impedance. The electrical impedance in each EEG sensor directly affects the signal-to-noise ratio, impacting the accuracy of neuronal pattern detection [25]. High impedance levels can generate artifacts and reduce the fidelity of the acquired signal, which, in turn, affects the performance of BCI classification algorithms.

Electrode impedance depends on the conductivity between the sensor and the subject’s scalp, skin preparation, the application of conductive gel, and contact pressure. Previous studies have shown that maintaining impedance below 5 kΩ improves EEG recording quality and reduces interference from environmental noise [26].

In the development of EEG classification models, it is crucial to consider strategies to mitigate the effects of impedance, such as sensor calibration, the use of robust signal processing techniques, and the optimization of electrode placement to minimize variations in recording quality.

To improve the efficiency and accuracy of MI-based BCIs, multiple studies have explored innovative approaches that include customization of system parameters and adaptation to each user’s neurophysiological characteristics [27,28]. Customizing BCI systems has proven to be an effective strategy to enhance performance, especially when considering factors such as attentional capacity, memory, and learning styles [29]. Additionally, identifying low-performing users in the initial stages of BCI training allows for interventions to optimize their performance in MI tasks [16,30].

From a technical perspective, classifying EEG signals in MI tasks poses challenges that require advanced signal processing and machine learning methods [31,32]. In this context, elastic net regression stands out as a powerful technique for addressing feature selection and regularization in high-dimensional and noisy datasets, such as those obtained from EEG [33]. By combining lasso and ridge penalties, elastic net handles multicollinearity and selects relevant features without incurring overfitting [34]. This approach improves the quality of the predicted signals, which is crucial for increasing the accuracy of BCI systems [35,36].

Another key method in EEG signal processing is the Common Spatial Pattern (CSP) [37], which maximizes the variance between different classes of EEG signals, facilitating the identification of the most relevant neural patterns for MI tasks [38]. CSP projects the data into a space where differences between brain activation states are more pronounced, improving class separability [39,40].

For accurate signal classification, Support Vector Machine (SVM) algorithms are particularly effective in BCI systems due to their ability to handle high-dimensional spaces and their skill in defining the optimal hyperplane that maximizes separation between classes [41]. Using nonlinear kernels [27,42], SVM captures complex patterns and ensures precise classification even in EEG data with nonlinear boundaries [43,44].

Recent research has incorporated advanced techniques, such as deep neural networks and regression algorithms [45], that improve the classification and prediction of MI signals [46]. These techniques not only allow for greater accuracy in interpreting EEG data but also facilitate the customization of BCIs to the individual characteristics of each user [47]. However, one of the main challenges in the practical implementation of BCIs remains the need to use a large number of electrodes to obtain high-quality EEG data [48,49]. This requirement poses a significant barrier in terms of comfort and practicality, especially for prolonged use [50].

In response to these limitations, our study proposes an innovative approach that employs signal prediction techniques to estimate EEG activity using a reduced number of electrodes. By applying elastic net regression, this method not only adapts to interindividual variability but also offers a practical solution to the traditional limitations of BCIs that require dense and costly electrode configurations. This proposal aligns with current trends in BCI research, which seek to develop more accessible and user-friendly technologies, facilitating their integration into everyday and clinical applications.

In recent years, advancements in signal processing and machine learning have allowed EEG-based BCIs to become more precise and accessible. However, for these technologies to be effectively integrated into daily life and clinical environments, systems must be efficient and comfortable for users. Recent research has emphasized the importance of developing BCIs with reduced electrode setups and lighter algorithms, which not only maintain or improve classification accuracy but also reduce cost and setup time [40]. Additionally, efforts to enhance model robustness against noisy and non-stationary EEG signals have gained attention, employing more advanced regularization and filtering techniques. This trend aligns with the shift towards portable, low-cost, and user-friendly BCIs, which could pave the way for broader adoption in neurological rehabilitation and other applications [39,46].

The rest of the article is organized as follows: Section 2 briefly reviews the theoretical background of the proposed model. Section 3 describes the experimental setup, including the dataset used. Section 4 presents the performance evaluation of the elastic net regression model network, describes the results, and discusses the findings. Finally, Section 5 provides critical insight into the performance provided and addresses some limitations and possibilities of the approach presented.

## 2. Materials and Methods

### 2.1. Signal Processing

Consider that *n* training trials are recorded, where the *i*-th trial is denoted as $x_{i}\in R^{v\times m}$, with *v* representing the number of channels and *m* the number of sampling points. The channels in the central region are a subset of the *v* channels. To develop the regression model, and, based on the time delay τ, the training data *X* are constructed as a concatenated matrix of all the *u*-channel EEG trials $\left(x_{i,u},i=1,\ldots ,n\right)$ along with their time-delay versions $\left({\tilde{x}}_{i,u},i=1,\ldots ,n\right)$ [51]:(1)$$X=\left[\begin{array}{ccccccc} x_{1,u} & x_{2,u} & x_{3,u} & x_{4,u} & x_{5,u} & \cdots & x_{n,u}\\ {\tilde{x}}_{1,u} & {\tilde{x}}_{2,u} & {\tilde{x}}_{3,u} & {\tilde{x}}_{4,u} & {\tilde{x}}_{5,u} & \cdots & {\tilde{x}}_{n,u} \end{array}\right]\in R^{2u\times n(m-\tau )}$$ where the parameter τ is defined as a positive integer $\tau \in Z^{+}$, as it represents a time delay in discrete sampling units. This ensures that it can be used unambiguously in the segmentation of EEG signals and in the mathematical formulation of the model.

The training data *Y* are formed by concatenating all the EEG trials from the *v*-channel, represented as(2)$$Y=\left[\begin{array}{cccc} x_{1,v} & x_{2,v} & \cdots & x_{n,v} \end{array}\right]\in R^{v\times n(m-\tau )}$$

In this expression, $x_{i,u}\in R^{u\times (m-\tau )}$ corresponds to the data from the *i*-th trial using *u*-channels, consisting of m−τ sampling points (from the first sampling point to the (m−τ)-th sampling point). Additionally, ${\tilde{x}}_{i,u}\in R^{u\times (m-\tau )}$ denotes the data from the *i*th trial using *u*-channels, with m−τ sampling points (ranging from the (τ+1)-th sampling point to the *m*-th sampling point). Lastly, $x_{i,v}\in R^{v\times (m-\tau )}$ indicates the data from the *i*-th trial using *v*-channels, which include m−τ sampling points (from the first sampling point to the (m−τ)th sampling point) [51]. The selection of the time delay τ in the regression model was not arbitrary but was conducted through an experimental evaluation to find the optimal value in terms of classification accuracy. Different values of τ were tested on the dataset. The results indicated that τ=1 provided the highest classification accuracy, preventing the loss of synchronization between the input signals *X* and the target signals *Y*. From a neurophysiological perspective, τ=1 allows capturing sufficient temporal context without losing samples within the observation window, improving prediction without affecting the temporal alignment of the signals.

Segmenting data into $x_{i,u}$, ${\tilde{x}}_{i,u}$, and $x_{i,v}$ is essential in multichannel EEG processing, as it supports the prediction and reconstruction of missing or noisy channels by leveraging information from adjacent segments. This technique enhances data quality by reducing noise and improving the signal-to-noise ratio (SNR), enabling more robust analyses for applications like motor imagery classification and cognitive state monitoring [52]. By introducing a time lag τ between segments, the model effectively captures temporal dependencies critical in time series analysis, especially within BCI systems, where tracking neural pattern evolution over time is crucial for accurate classification and prediction [53].

Additionally, variables such as $x_{i,u}$ and ${\tilde{x}}_{i,u}$ enhance feature extraction in predictive models, including convolutional neural networks (CNNs) and recurrent neural networks (RNNs). These models exploit the spatial and temporal dependencies in segmented EEG data to identify and classify patterns in motor imagery tasks more effectively [53]. The representation of EEG data across multiple channels and time points further facilitates the use of regularization techniques, such as elastic net and lasso regression, refining the feature space by addressing multicollinearity and noise. This structured approach to segmentation and processing improves model performance, supporting higher accuracy and reliability in real-time BCI applications [54,55].

### 2.2. Elastic Net

For a given dataset ${\left\{\left(x_{i},y_{i}\right)\right\}}_{i=1}^{n}$ where $y_{i}\in R$, we consider a simple linear regression model:(3)y=Xβ+ε
where $y={\left(y_{1},y_{2},\ldots ,y_{n}\right)}^{T}$ represents the response variable and $X={\left(x_{1},x_{2},\ldots ,x_{p}\right)}^{T}$ is the full rank design matrix. In this context, $x_{j}={\left(x_{1j},x_{2j},\ldots ,x_{nj}\right)}^{T}$, for j=1,2,…,p denotes the *p*-dimensional explanatory variable. Additionally, $\beta ={\left(\beta_{1},\beta_{2},\ldots ,\beta_{p}\right)}^{T}$ refers to the associated vector of regression coefficients, and ε is the vector of i.i.d. random errors with a mean of zero. Without loss of generality, we can assume that the response variable is centered and that the predictors have been standardized following a location and scale transformation [56].(4)$$\sum_{i=1}^{n}y_{i}=0,\;\sum_{i=1}^{n}x_{ij}=0,\;\sum_{i=1}^{n}x_{ij}^{2}=1,\;j=1,2,3,4\ldots ,p.$$

Nevertheless, if *X* does not have full column rank, or if there is a significant linear correlation among some columns, the determinant of $X^{T}X$ approaches 0, meaning $X^{T}X$ is nearly singular. The conventional OLS method may lack stability and reliability in these cases. To address this issue, Hoerl and Kennard [56,57] proposed ridge regression:(5)$$\mathrm{Ridge}\;\mathrm{Regression}\text{:}\;L\left(\beta \right)=\sum_{i=1}^{n}{\left(y_{i}-x_{i}\beta \right)}^{2}+\lambda \sum_{j=1}^{p}\beta_{j}^{2}.$$

This penalty approach enhances OLS by converting the unfit problem into a fitting problem. Although it sacrifices the unbiased nature of OLS in exchange for improved numerical stability, it yields greater computational accuracy. While ridge regression effectively mitigates the issues arising from high correlation between variables and enhances prediction accuracy, it does not suffice for model selection on its own. Consequently, Tibshirani [56,58] introduced the following primary lasso criterion:(6)$$\mathrm{Lasso}\text{:}\;L\left(\beta \right)=\sum_{i=1}^{n}{\left(y_{i}-x_{i}\beta \right)}^{2}+\lambda \sum_{j=1}^{p}\left|\beta_{j}\right|.$$
where λ>0 is a constant adjustment parameter. Lasso is a method that applies penalization to ordinary least squares. Due to the singularity of the derivative of the penalty function at zero, the coefficients of non-significant variables are driven to zero, while a smaller compression is applied to the significant independent variables with larger estimates, ensuring accuracy in the parameter estimations.

Nonetheless, lasso possesses some intrinsic limitations: it lacks the Oracle property and has the drawback of selecting at most *n* variables when the sample size (p>n) is considered. When multiple characteristics are correlated, lasso tends to choose only one among them. Furthermore, lasso is less effective than ridge regression when dealing with independent variables that exhibit multicollinearity. In light of these issues, Zou and Hastie introduced the elastic net [56]:(7)$$\mathrm{Elastic}\;\mathrm{Net}\text{:}\;L\left(\lambda_{1},\lambda_{2},\beta \right)=\sum_{i=1}^{n}{\left(y_{i}-x_{i}\beta \right)}^{2}+\lambda_{2}\sum_{j=1}^{p}\beta_{j}^{2}+\lambda_{1}\sum_{j=1}^{p}\left|\beta_{j}\right|.$$

The elastic net employs both the $\ell_{1}$ and $\ell_{2}$ norms within linear regression models that incorporate prior canonical terms. It merges the benefits of both lasso and ridge regression. This method addresses the challenge of variable selection in the presence of unknown groupings. When compared to the lasso, the elastic net also enhances the handling of data with sample sizes (p>n) and variables with multicollinearity. Unfortunately, it should be noted that the elastic net cannot completely eliminate the effects caused by noisy data [56].

### 2.3. Common Spatial Pattern (CSP)

At the second stage, band-pass-filtered signals are spatially processed using the Common Spatial Patterns algorithm. The CSP algorithm is applied to spatially filter EEG signals into a low-dimensional space through linear transformations. The primary goal of CSP is to minimize the variance within two classes while maximizing the variance between them. The spatial filters generated by CSP are commonly utilized for detecting Event-Related Desynchronization (ERD) and Event-Related Synchronization (ERS). However, a limitation of the CSP algorithm is its ability to discriminate only between two classes.

A motor imagery trial can be represented as $X\in R^{N\times T}$, where *N* denotes the number of channels and *T* is the number of time samples. The projection matrix $W\in R^{N\times N}$, obtained through the CSP algorithm, projects the trial data *X* to the source signals $S\in R^{N\times T}$, as shown below [59]:(8)S=WX

In this case, the rows of the projection matrix *W* correspond to spatial filters. A subset of these spatial filters from *W* is relevant for classification tasks. Once the matrix *S* is determined, the rows of *S* can be used for classification purposes.

The rows of matrix *S*, corresponding to the first and last *m* eigenvalues sorted in descending order, are selected for further analysis. The feature related to each selected row of matrix *S* is calculated as follows:(9)$$f_{q}=\log\left(\frac{\mathrm{var}(s_{q})}{\frac{1}{2m}\sum_{i=1}^{2m}\mathrm{var}\left(s_{i}\right)}\right),\;q\in \left\{1,\ldots ,2m\right\}$$
where $\mathrm{var}(s_{q})$ represents the variance of the *q*-th row of the selected rows from matrix *S*.

The spatial filters are obtained by applying the One-Versus-Rest (OVR) CSP algorithm for a four-class motor imagery-based BCI system. Specifically, the CSP algorithm is executed four times, and, in each instance, the top two spatial filters are chosen. Consequently, for the four-class classification, eight features are extracted through the repetition of the CSP algorithm for each class. Finally, the features from each trial of each class are compiled within a feature vector containing all the windows across the nine sub-bands [59].

### 2.4. Support Vector Machines (SVMs)

The Support Vector Machine was introduced by Vladimir Vapnik [60]. The closest sample to the hyperplane is referred to as a support vector, and the distance between two support vectors defines the margin. Maximizing the margin between two classes enhances separability, as seen in Figure 1.

Thus, the primary objective of SVM is to find the hyperplane that provides the largest possible margin. The general form of the hyperplane can be expressed as(10)$$g\left(x\right)=w^{T}x+w_{0}$$
where x represents the feature vector of a sample, and w is the normal vector to the hyperplane. The hyperplane divides the feature vectors into distinct classes. This property is formalized as(11)$$\left\{\begin{array}{c} g(x)\geq 1\;\mathrm{for}\;\mathrm{class}1\\ g(x)\leq -1\;\mathrm{for}\;\mathrm{class}2 \end{array}\right.$$

The distance from the hyperplane to the nearest support vector is calculated as(12)$$z=\frac{\left|g(x)\right|}{∥w∥}=\frac{1}{∥w∥}$$

Thus, the margin is given by(13)$$\frac{1}{∥w∥}+\frac{1}{∥w∥}=\frac{2}{∥w∥}$$

The objective of SVM is to minimize the normalized weight vector ∥w∥, allowing for the maximization of the margin. The minimization of ∥w∥ constitutes a nonlinear optimization problem, which is addressed using the Karush–Kuhn–Tucker (KKT) conditions. By applying Lagrange multipliers $a_{i}$, three KKT conditions can be expressed as follows:(14)$$w=\sum_{i=0}^{N}a_{i}y_{i}x_{i}$$(15)$$\sum_{i=0}^{N}a_{i}y_{i}=0$$(16)$$a_{i}\geq 0,\;i=1,\ldots ,N$$
where *y* takes the values −1 or 1, indicating the class label of the samples. The following cost function is maximized to obtain w, which optimizes the margin [61]:(17)$$\tilde{L}\left(a\right)=\sum_{i=0}^{N}a_{i}-\frac{1}{2}\sum_{i=1}^{N}\sum_{j=1}^{N}a_{i}a_{j}y_{i}y_{j}x_{i}^{T}x_{j}$$

## 3. Experimental Setup

In this study, we worked with the BCI IIa dataset, which contains EEG signals from 9 subjects performing motor imagery tasks focused on the left hand and the right hand. The primary objective was to develop a method to improve the classification of these signals using a prediction and classification approach based on regression and spatial pattern analysis, following the methodology outlined in Figure 2.

A multiple linear regression (MLR) model was implemented to estimate EEG signals in a greater number of channels from a subset of input channels. To improve the coefficient of determination r^2^, an elastic net-based regressor was proposed, which yielded enhanced results compared to the original MLR model. This model was designed to increase the amount of MI-related information available for classification. During the training phase, the elastic net model was built using training data consisting of EEG signals recorded from a large number of electrodes and a small subset of them. Subsequently, the regression model predicts EEG signals in all channels using only the data from a reduced subset of channels. This allows the reconstruction of signals across all channels from a limited set of electrodes.

Once the EEG signals were predicted by the regularization model, the CSP method was employed to extract the most relevant features related to MI tasks from the complete signals. The CSP model was adjusted using the signals predicted by the regression model to maximize the variance between MI classes (left hand and right hand). These extracted features were then used as input for the classifier.

For the classification of MI signals, an SVM classifier was employed. The aim was to differentiate between motor imagery tasks of the left hand and the right hand.

During the training phase, the SVM classifier was trained using the features extracted by the CSP model from the complete EEG signals. With this information, the confusion matrix was generated, resulting in an effective classification of MI tasks. The average accuracy, considering all 9 subjects and the 22 channels, was 78.16%.

### 3.1. BCI Competition IV Dataset IIa

The dataset used in this study is available at https://www.bbci.de/competition/iv/ accessed on 1 February 2025. It is publicly accessible and can be used by the research community. Originally, it was published by Brunner et al. (2008) and is available for download on the official BCI Competition platform [62].

This dataset contains EEG signals from 9 subjects performing four motor imagery tasks: left hand, right hand, both feet, and tongue. Subjects have been assembled according to the experimental paradigm for MI, as shown in Figure 3. For each subject, two sessions were recorded on different days, each consisting of 6 runs with 48 trials per run (12 trials per task), totaling 288 trials per session. During the trials, an arrow was displayed on the screen to indicate the motor imagery task (left hand, right hand, feet, or tongue), and subjects performed the task without feedback. The EEG signals were recorded from 22 channels at a sampling rate of 250 Hz. For this study, the EEG data from two motor imagery classes (left hand vs. right hand) were used for classification.

The EEG was recorded using 22 Ag/AgCl electrodes spaced 3.5 cm apart, in a monopolar setup with the left mastoid as the reference and the right mastoid as the ground, as shown in Figure 4. The signals were sampled at 250 Hz, filtered with a band-pass between 0.5 Hz and 100 Hz, and a 50 Hz notch filter was applied to eliminate line noise. The amplifier sensitivity was set to 100 µV.

### 3.2. Data Setup and Preprocessing

At this stage of the experiment, 8 specific EEG channels were selected to predict activity across the 22 available channels in the dataset. The selected channels (C3, C1, Cz, C2, C4, CP1, CPz, and CP2) correspond to areas of the scalp highly related to motor control, as shown in Figure 5, being especially relevant in tasks of motor imagery classification. This selection allows for a reduction in data dimensionality without losing crucial information. The goal of this stage is to use the information captured by these 8 channels to predict the complete signals in the 22 channels (Fz, FC3, FC1, FCz, FC2, FC4, C5, C3, C1, Cz, C2, C4, C6, CP3, CP1, CPz, CP2, CP4, P1, Pz, P2, and POz), applying regression techniques and spatial pattern analysis. This approach optimizes computational processing and facilitates the interpretation of brain behavior, while preserving the ability to adequately model the neural dynamics recorded across all available electrodes.

The data processing was conducted through the following process, generally applicable to multiple subjects within the dataset: EEG data were extracted from the subjects, and preprocessing techniques were applied to optimize the extraction of relevant features for analysis. Among these techniques, EEG sensor selection and a band-pass filter between 8 and 30 Hz were included, designed to remove noise components or those unrelated to the signals of interest.

Subsequently, the EEG data were segmented into temporal windows according to a method specifically defined for this study. The windows had a duration of 6.8 s, with a start offset of −1.9 s and an end offset of 2 s. These configurations allowed for the precise capture of motor signals corresponding to different movement classes, which, in this case study, focused on imagined movements of the right hand and the left hand. Additionally, the duration of the imagined movement was set at 3 s, and no overlap between windows was applied, with a value of 0.0. A summarized version of this preprocess can be seen in Figure 6.

Features were specifically extracted for the classes corresponding to left- and right-hand movements. These EEG signals, differentiated by motor classes, were then used as the basis for classification analysis. To ensure proper model evaluation, the data were split into training and test sets in a 70/30 ratio, allowing for a robust assessment of the classification model’s performance.

The CSP technique was employed to reduce the data dimensionality and enhance discrimination between the motor classes. This approach captures the most significant and important differences in signals between the study’s classes. After transforming the data with CSP, a linear kernel SVM classifier was applied to classify the motor signals. The model’s performance was evaluated using the coefficient of determination r^2^ and overall accuracy. The model’s accuracy, evaluated through the confusion matrix and classification metrics, demonstrated a high capacity for discriminating between left- and right-hand movement classes. Overall, the CSP- and SVM-based model showed considerable accuracy in classifying motor signals across multiple subjects.

## 4. Results

### 4.1. Elastic Net Regularization Results

This section presents a comparison between the recorded EEG signals (in blue) and the predicted signals (in red) for several different channels from a total of 22 channels; see Figure 7. The results were obtained using the elastic net regression model with the following parameters: alpha = 1, l1_ratio = 0.5, max_iter = 1000, and random_state = 42. This model combines L1 (lasso) and L2 (ridge) regularizations, allowing it to effectively handle multicollinearity and select relevant features, optimizing the signal prediction. To identify optimal hyperparameters for the elastic net model, tuning was performed based on cross-validation using ElasticNetCV from Scikit-Learn. Different values of alpha and l1_ratio were explored, where the optimal combination obtained was alpha = 1 and l1_ratio=0.5. The value alpha = 1 provides moderate regularization, allowing a balance between bias and variance. Meanwhile, l1_ratio=0.5 balances lasso and ridge regularization, optimizing feature selection without compromising useful information from EEG channels.

From the predicted dataset, three channels from subject 9 were plotted to observe the result of the elastic net regularization. In Figure 7, it can be seen that the model adequately captures the main characteristics of the original signals in the five channels shown: FCz, FC2, and FC1. The predictions closely follow the recorded signals, especially in terms of phase and frequency, which indicates that the model has been able to correctly generalize the neural activation patterns from the eight selected input channels.

The model performs well across most samples, maintaining a visual correlation between the recorded signal and the predicted signal. However, slight deviations in amplitude can be observed at certain points, especially at peaks of higher magnitude, which could be attributed to the nature of elastic net regularization.

Figure 8 shows the regularization results for subject 2 in three channels: FCz, FC2, and FC1. The prediction model used demonstrates a high degree of agreement between the real and predicted signals, indicating an accurate estimation of brain activity in these channels. The response is particularly notable in the amplitudes and oscillations over time, although small discrepancies can be observed in the peaks of some samples.

To obtain the results of the regularization model, the coefficient of determination was calculated for the nine subjects, as shown in Table 1. The regression model demonstrates consistent performance, with r^2^ values ranging between 0.567 and 0.641, indicating that the model is capable of capturing a considerable proportion of the variability in the original data, with an acceptable level of fit for all subjects. Subject 2 shows the highest r^2^ value (0.641), suggesting that, in this case, the model is particularly effective in predicting the signals, while subjects 6 and 7 show the lowest value (0.567), which still represents a reasonable fit, although with a slight loss of accuracy compared to the other subjects.

### 4.2. Model Results for Common Spatial Patterns

After applying the regularization model, the Common Spatial Patterns of the nine subjects were calculated using six CSP components, as shown in Figure 9. These patterns reflect the spatial distribution of neural activity based on motor imagery and are essential for feature extraction, enabling the discrimination between different motor classes, such as left- and right-hand movements. The analysis of each subject reveals differences in the distribution of the patterns, suggesting inter-subject variability in cortical activations during MI tasks.

The results for subject 1 show a relatively balanced distribution between positive and negative patterns in the CSP components, with higher activity concentrated in the lateral areas of the scalp. This suggests bilateral activation of the motor regions during the motor imagery task. Subject 2, on the other hand, presents a strong positive activation in CSP0 in the frontal and lateral regions, which could indicate more pronounced lateralization in motor tasks.

Subject 3 exhibits strong activation in CSP5, highlighting a concentration in the parietal areas, suggesting a significant contribution of these regions to imagined motor control. In contrast, subject 4 shows a more evenly distributed activation across different components, indicating the involvement of multiple cortical areas without pronounced lateralization.

For subject 5, strong activations are noted in both CSP0 and CSP4, which could indicate a high correlation of signals in these areas during the motor imagery task. Subject 6 shows a more diffuse distribution of patterns, possibly reflecting greater noise or variability in the EEG signals for this particular subject.

Subject 7 exhibits strong negative activation in the lateral areas in CSP0, suggesting clear lateralization of signals related to the motor task. Subject 8, in turn, shows more centralized activation in CSP1 and CSP3, possibly related to a more diffuse activation of the primary and secondary motor areas.

Finally, subject 9 displays a relatively symmetrical pattern in CSP0 to CSP2, with moderate activations in the central and lateral regions. This indicates that the signals from this subject show a good correlation between cortical areas involved in the motor imagery task, without extreme lateralization. The results obtained from the CSP model are summarized in Table 2.

### 4.3. Classification Model

Figure 10 presents the confusion matrices obtained for the nine subjects in the MI classification process using an SVM model with features extracted from the CSP model. These matrices provide a clear representation of the performance of the classification model.

Table 3 shows the percentage of correct responses and the Kappa coefficient of the SVM model for each subject in motor imagery classification tasks. On average, the model achieved an accuracy of 78.16% and a Kappa of 0.56, indicating moderate performance in class discrimination. Notably, subjects 4 and 8 achieved the highest accuracies with 95.24% and 90.16%, and Kappa values of 0.891 and 0.801, respectively, suggesting high consistency in classification for these individuals. In contrast, subjects 1 and 5 had the lowest accuracies (65.57% and 62.30%) and the lowest Kappa values (0.313 and 0.242), demonstrating the difficulties of the model in effectively generalizing in these cases. The inter/intra-subject variability present in the recorded EEG data has a significant impact on the individual performance achieved in MI task classification. This variability may be influenced by neurophysiological factors, such as differences in motor area activation and brain functional connectivity, as well as aspects related to the quality of the acquired EEG signal. The present study not only evaluates the model’s ability to discriminate between different MI tasks but also analyzes the influence of this variability on the accuracy achieved by each subject. In particular, it highlights the need to interpret how the brain generates mental responses with varying levels of consistency among subjects, which may explain the observed differences in the results. These findings suggest that future research could focus on developing model adaptation strategies tailored to individual characteristics to improve classification accuracy in subjects with lower performance.

### 4.4. Performance of the Classification Model with Respect to MI Tasks

Table 4 presents the performance results of the SVM classification model with respect to MI tasks for each subject, distinguishing between left-hand (class 1) and right-hand (class 2) tasks. For class 1, the model achieved an average accuracy of 0.785, a recall of 0.762, and an f1-score of 0.770. In class 2, the values were slightly higher, with an average accuracy of 0.785, recall of 0.797, and f1-score of 0.801, suggesting better performance in classifying this class compared to class 1.

Analyzing the subjects individually, it is observed that some, such as subject 4 and subject 8, achieve notable performance across all metrics, with precision and recall values above 0.90, indicating high accuracy in classifying both MI classes in these subjects. However, subjects like subject 1 and subject 5 present lower precision and f1-score values, which could indicate specific challenges for the model in adequately generalizing in these cases.

The average support is 28.77 for class 1 and 30.11 for class 2, reflecting a reasonable balance in the number of samples analyzed for each class, providing a fair evaluation in both MI tasks.

## 5. Discussion and Concluding Remarks

The development of BCI using MI as a control paradigm has proven promising for a variety of assistive and rehabilitation applications. However, variability in control capability among users, caused by interindividual differences in brain structure and function, remains a significant barrier to the widespread adoption of these technologies. Throughout this study, we have integrated advanced signal processing and machine learning techniques to address these challenges. The application of deep neural networks and regression algorithms, such as elastic net regression, has allowed not only a more accurate interpretation of EEG data but also a more effective customization of BCIs to fit the neurophysiological characteristics of individual users.

The need for multiple electrodes, which has traditionally been an impediment to practical and accessible BCI systems, was addressed by implementing a signal prediction model. This model has proven capable of estimating the necessary EEG activity for BCI operations with a reduced number of electrodes, maintaining data quality, and reducing both the complexity and cost of the system. However, after the evaluation stage, it is worth mentioning the following points.

*EEG Signal Processing and Filtering*: Band-pass filters with cutoff frequencies from 8 to 30 Hz were applied, allowing the EEG signals to focus on the bands most relevant for MI tasks, eliminating low- and high-frequency noise. At the same time, techniques for creating temporal windows were used to adjust the data according to motor tasks to improve feature extraction. This process is key for reducing the number of necessary electrodes, as it facilitates the prediction of the channels, as proposed in the study. The use of optimized windows and carefully calculated offsets, along with the elastic net regression model, ensures that the predicted signals can be efficiently used in MI classification, achieving a proper balance between accuracy and simplicity. This approach, implemented in this study, supports the feasibility of reducing the complexity and cost of EEG setups without compromising the quality of the classification.

*Accuracy Achieved by the Elastic Net Regularization Model*: The accuracy achieved by the coefficients of determination and the corresponding outcome of the EEG signals recorded and estimated by the 22 channels provides a critical perspective on the effectiveness of the elastic net model in predicting EEG signals from a limited number of channels. The analysis reveals that the determination coefficients r^2^ varied between 0.567 and 0.641 for different subjects, with an average of 0.587. This indicates a moderate to good correlation between the predicted and recorded EEG signals, which is a positive indication that the model can capture the underlying dynamics of the EEG signals with a reasonable degree of accuracy. Variability in model performance among different subjects, as reflected in the determination coefficients, can be attributed to individual differences in brain physiology and EEG signal characteristics. These differences underscore the importance of customizing prediction models or adjusting model parameters for each subject in practical applications, which could significantly improve the overall accuracy of MI-based BCIs. Results from the predicted and recorded EEG channels visually illustrate the similarity between the two EEG signals. The close correspondence in amplitude and waveform across the sampling points suggests that, despite the reduction in the number of channels used for the model input, the integrity of the information is largely maintained. This is crucial for practical applications where portability and the reduction of hardware complexity may be critical.

*Outcome from Common Spatial Patterns Model*: The CSP results reflect high inter-subject variability in the distribution of activation patterns. For example, in subjects 1, 3, 5, and 7, pronounced lateral activations were observed, suggesting clear involvement of the motor areas during the MI task. In contrast, some subjects, like subject 2, exhibited more diffuse and central activations, which may indicate differences in how individuals process the motor task. The amplitude and distribution of the patterns also vary considerably among the different CSP components. While some components showed a more focused distribution (e.g., CSP0 and CSP1 in several subjects), others exhibited more diffuse and bilateral patterns (e.g., CSP4 in subject 5). This suggests that the different CSP components capture distinct aspects of neural activation related to MI tasks. The analysis of CSP in MI tasks reveals brain activation patterns that varied significantly among subjects, highlighting the individual nature of neural responses during these tasks. While some subjects exhibited bilateral or uniform activations that could facilitate the classification of signals, others showed lateralized activations or were focused on specific areas, such as the motor and parietal regions, which are key for differentiating MI classes. This variability suggests that classification models should be individually adapted to maximize accuracy, leveraging the areas of highest brain activation for each subject [63].

In practical terms, the motor and parietal areas appear to be the most relevant for classification in most subjects, implying that BCIs could benefit from focusing feature extraction in these regions. However, the presence of diffuse activations in some subjects indicates that the model’s generalization capability is also important. A balanced approach that combines customization and robustness in feature extraction, based on observed variability, is essential for enhancing the effectiveness of BCI systems in practical applications.

*MI Classification*: Accuracy values ranged from 62.30% for subject 5 to 95.24% for subject 4, with an overall average of 78.16%. This indicates that, while some subjects could be classified with high accuracy, others presented greater challenges, possibly due to differences in signal quality or in the consistency of task execution. The Kappa coefficients, which adjust accuracy considering the possibility of random hits, also showed a wide range from 0.242 to 0.891, suggesting variations in the consistency of classifications among subjects. The variability in accuracy and Kappa values highlights the importance of personalizing BCI models. Adjusting model parameters or using more flexible and adaptive learning approaches could improve accuracy for subjects with lower performance [64]. The high accuracy and Kappa values in some subjects demonstrate the potential of the elastic net approach combined with SVM to provide precise classifications in BCI setups, especially when handling complex multidimensional data like EEG. The proposed method outperforms traditional approaches that use fewer electrodes and improves performance compared to full-channel EEG-based methods. Table 5 presents a comparison, highlighting the number of channels used, the methodology employed, and the accuracy achieved in various studies.

The present method, based on elastic net and using only eight channels, achieves competitive accuracy compared to traditional and deep learning approaches that employ 22 channels. Although some deep learning methods may achieve slightly higher accuracy, they require more data and higher computational costs. In contrast, our approach optimizes channel selection, reducing hardware complexity and avoiding information redundancy, making it more efficient and practical for real-world applications.

*CSP and SVM*: While CSP/SVM is a standard method in BCI, it is acknowledged that comparing it with deep learning models can provide better contextualization of performance. Models such as CNNs, RNNs, and Transformers have proven to be effective in EEG signal classification, particularly in motor imagery tasks. Table 6 presents the advantages and disadvantages of these models.

The CSP + SVM method used in this study offers a computationally efficient and interpretable approach for EEG-MI signal classification, making it particularly suitable for small datasets. Unlike deep learning models such as CNNs, RNNs, and Transformers, which require tuning multiple hyperparameters (e.g., learning rate, batch size, number of layers, and dropout rates), CSP + SVM relies on a smaller set of critical hyperparameters, such as the regularization parameter (C), kernel type, and gamma value. While deep learning models have the advantage of automatically extracting spatial and temporal features from raw EEG signals, they often require large volumes of data and high computational power, making them less practical for real-time applications. CSP + SVM, despite its limitation in capturing complex nonlinear relationships, remains an effective option for motor imagery classification due to its lower computational cost and robust feature extraction capability when applied to well-defined frequency bands.

In recent years, deep learning-based methods, especially CNNs, have demonstrated outstanding performance in EEG signal classification, including motor imagery (MI) tasks. Recent studies have used CNNs to automatically extract spatial and temporal features, achieving improvements in classification accuracy [72,73]. In particular, Zhang et al. (2021) [72] reported an accuracy of 88.4% in MI classification using a deep neural network architecture called EEG-Inception, compared to traditional methods such as CSP and SVM. Another relevant study, Gallo and Phung (2022) [73], developed a CNN-based model that achieved high accuracy across multiple BCI paradigms, with improvements in generalization capability across different subjects and sessions. In our study, the elastic net-based model achieved an average accuracy of 78.16%, with variability ranging from 62.30% to 95.24% depending on the subject. While this accuracy is competitive, deep learning models have shown advantages in capturing complex patterns in EEG signals, which can enhance the system’s robustness under different experimental conditions. However, it is important to consider differences in the amount of data and computational resources used in these studies.

Unlike CNNs, elastic net is a linear regularization method that offers significant advantages in scenarios where available data are limited.
Elastic net offers several advantages for EEG signal classification. It requires significantly fewer computational resources compared to CNNs, facilitating its implementation in real-time applications or hardware-constrained environments. Additionally, it provides interpretability, as it allows identifying which features contribute to classification, whereas deep learning models often act as a black box. Another key benefit is its ability to reduce overfitting; by combining L1 and L2 regularization, it effectively handles multicollinearity in EEG data, which is critical in scenarios with limited data [74]. However, elastic net also presents certain limitations. Its modeling capacity is constrained, as it does not capture complex nonlinear relationships in EEG signals, which may restrict its performance compared to nonlinear methods such as CNNs and RNNs. Furthermore, unlike CNNs, which learn representations directly from raw data, elastic net relies on the manual selection of relevant features, making its generalization potentially more dependent on feature engineering.

Despite the results obtained, the study presents certain limitations that may affect its applicability. The feature selection based on central EEG channels could restrict its use in configurations with fewer electrodes or different spatial distributions.

This study has demonstrated that optimization in the design of BCIs can significantly enhance the accessibility and practicality of these technologies without sacrificing performance. By reducing the number of electrodes and implementing advanced techniques for EEG signal processing, the integration of BCIs into both clinical and home environments has been facilitated, thus broadening their potential applications. The application of prediction methods based on reduced regression models has validated the effectiveness of these techniques in accurately estimating EEG signals [1]. These methods have not only proven capable of revolutionizing BCI design due to their accessibility and efficiency, but they have also highlighted the importance of adapting the models to the individual characteristics of each user to maximize accuracy and effectiveness in classifying motor imagery tasks. The results of the study illustrate the robustness of the CSP approach in discriminating MI tasks, emphasizing the need to consider individual variability in the distribution of activation patterns. This customization is crucial for optimizing the performance of BCI systems, given that CSP patterns are essential for the accurate classification of EEG signals. The overall approach has proven effective; customization and optimization of the models are fundamental to overcoming individual limitations and enhancing the functionality of BCIs in practical applications. Future research is recommended to focus on developing adaptive methods that dynamically adjust to the specific neural characteristics of each user, thus ensuring the clinical relevance and utility of BCIs in a variety of contexts.

For future work, the authors plan to explore the use of different sets of channels beyond the motor area, as well as the implementation of deep learning models for signal prediction. In this regard, the application of EEGNet is being evaluated to improve the spatial representation of signals, along with the use of Variational Autoencoders to optimize prediction. Additionally, future research should focus on the continuous optimization of prediction and classification algorithms to enhance the robustness and accuracy of MI-based BCIs. It would also be essential to delve deeper into the impact of individual differences in brain anatomy and physiology on BCI interaction, which would enable the development of faster and more personalized calibration protocols. This would contribute to improving user experience and the effectiveness of rehabilitation applications. 

## Figures and Tables

**Figure 1 sensors-25-02259-f001:**
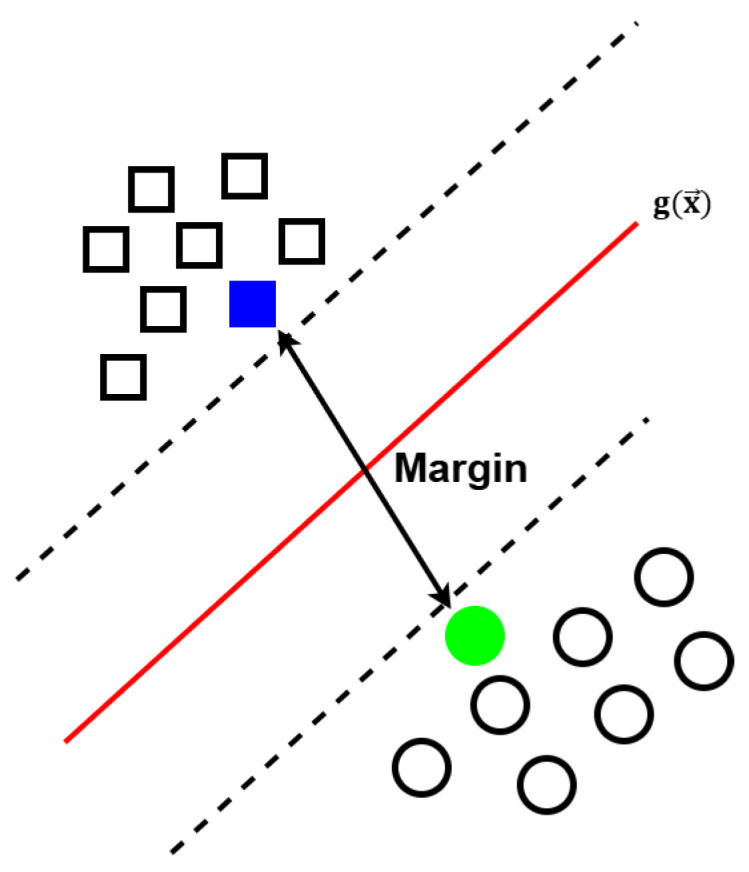
A simplified illustration of SVM, where the filled shapes denote the support vectors. The margin refers to the distance between these support vectors [61].

**Figure 2 sensors-25-02259-f002:**
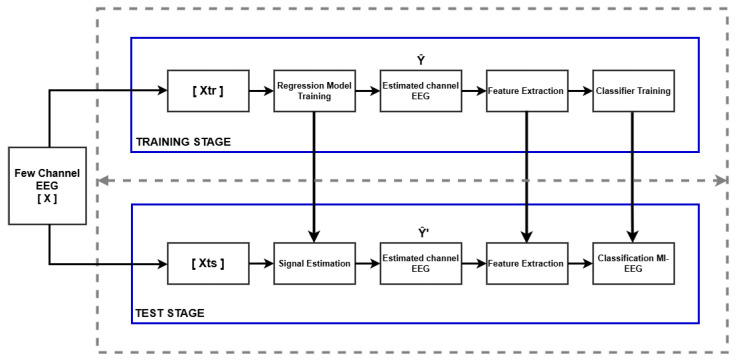
Guideline of the proposed framework to improve the classification of these signals using a prediction and classification approach based on regression and spatial pattern analysis.

**Figure 3 sensors-25-02259-f003:**
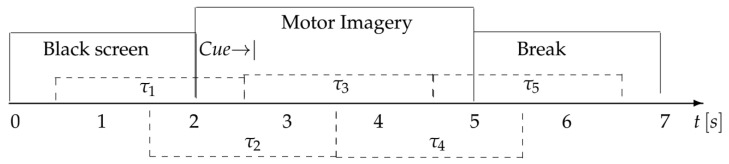
Timeline of the BCI Competition IV Dataset IIa database, of the motor imagery paradigm evaluated.

**Figure 4 sensors-25-02259-f004:**
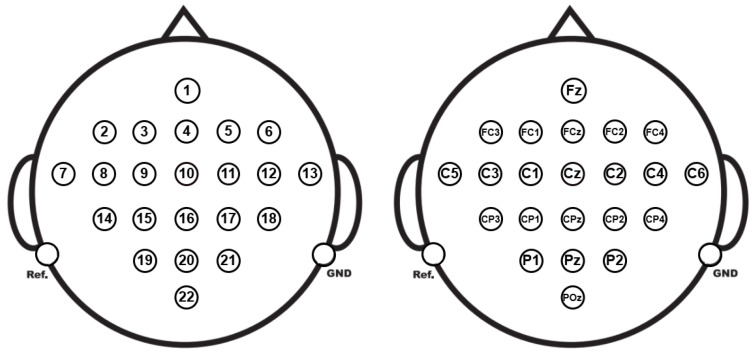
Electrode montage corresponding to the international 10–20 system. EEG channel configuration: numbering (**left**) and corresponding labels (**right**).

**Figure 5 sensors-25-02259-f005:**
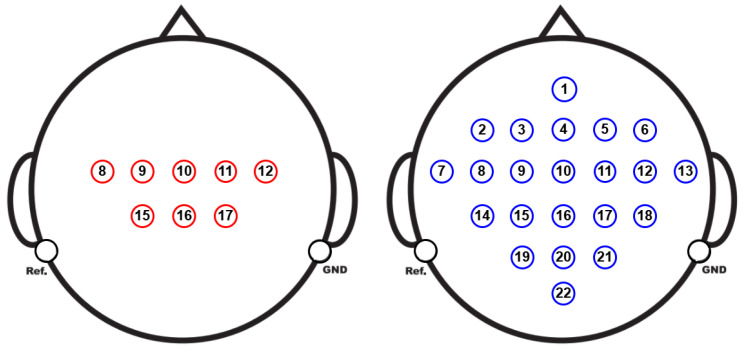
Position of the 8 strategically selected channels located in cortical areas highly related to MI. These positions correspond to scalp regions covering primary motor and premotor areas, such as the somatosensory cortex, which are essential for the neuronal representation of imagined movements (**left**). Extended configuration of 22 channels, predicted and extrapolated from the initial 8 channels (**right**).

**Figure 6 sensors-25-02259-f006:**
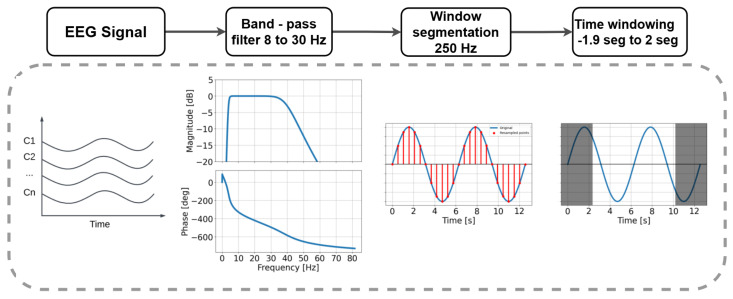
Scheme for preprocessing the EEG input signal for MI-EEG classification.

**Figure 7 sensors-25-02259-f007:**
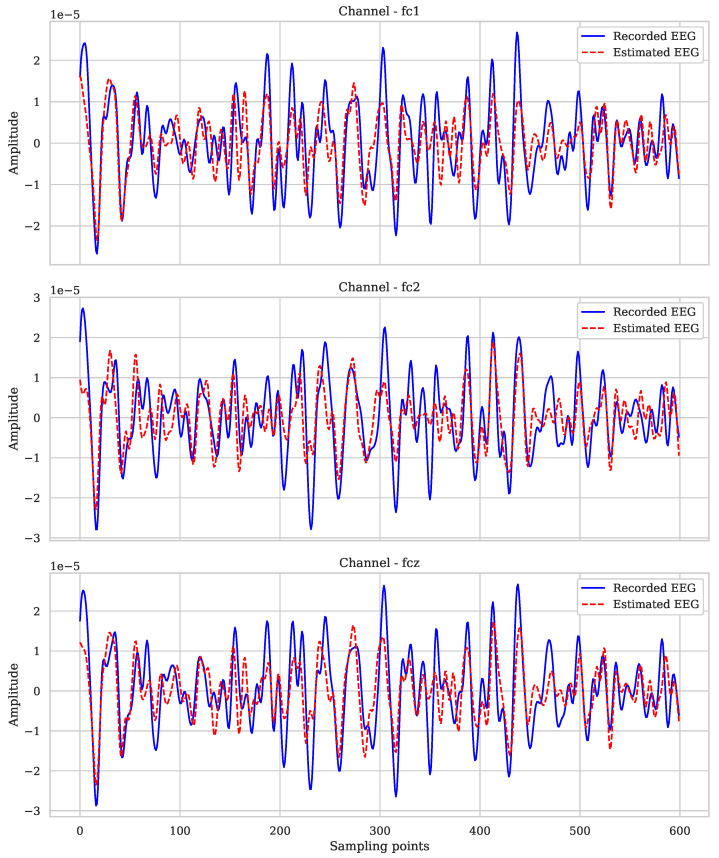
Comparison between the recorded EEG and the estimated EEG of subject 9.

**Figure 8 sensors-25-02259-f008:**
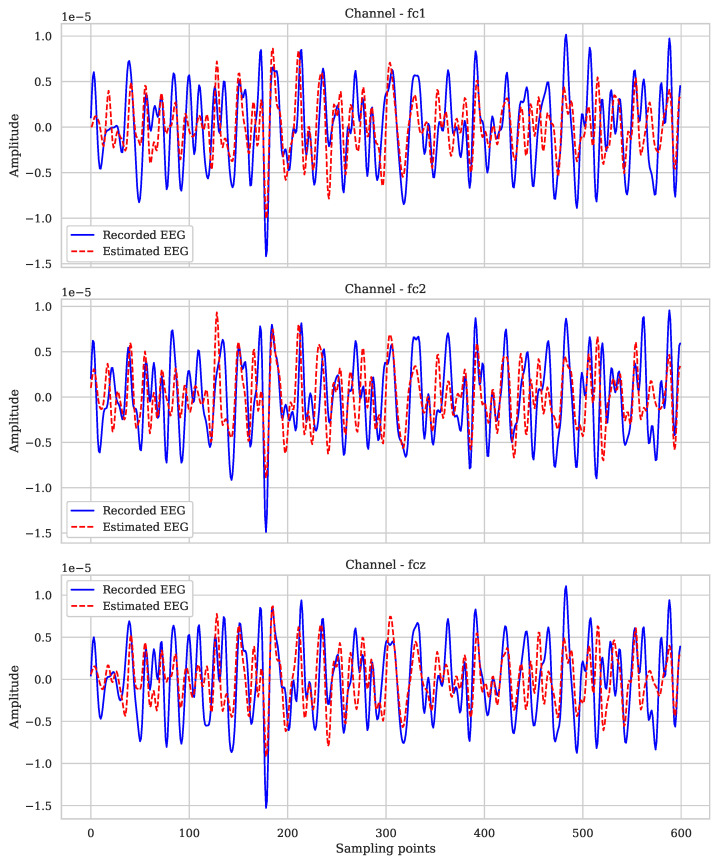
Comparison between the recorded EEG and the estimated EEG of subject 2.

**Figure 9 sensors-25-02259-f009:**
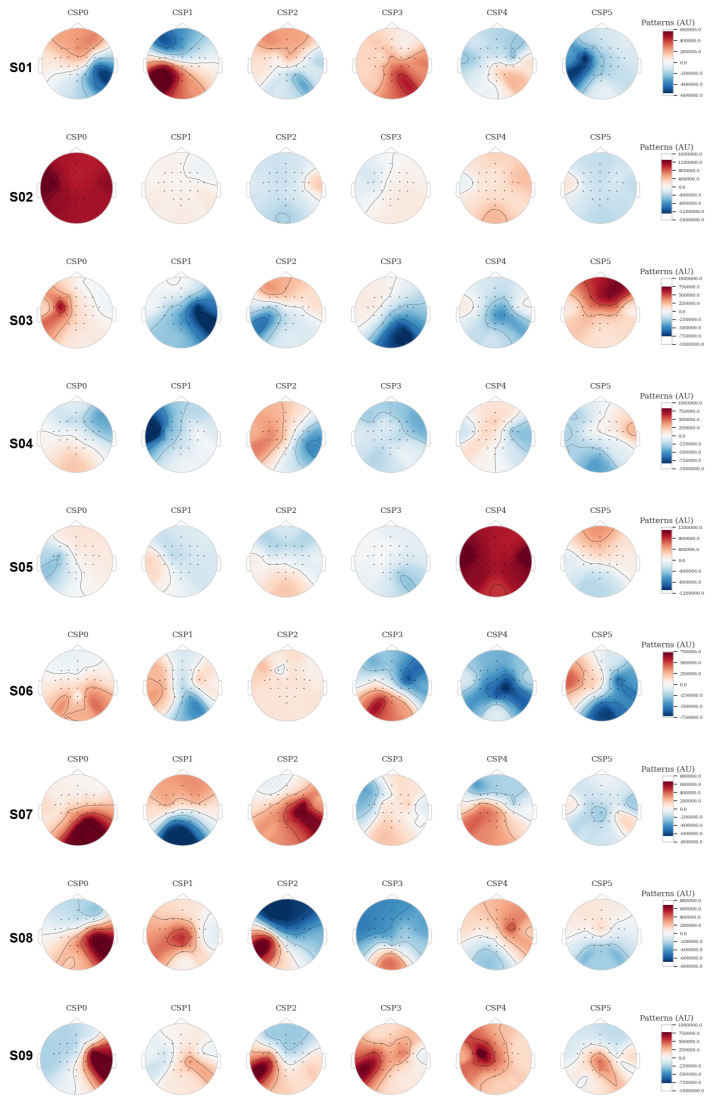
EEG topoplot of the CSP components of the subjects. These maps represent the spatial distribution of neuronal activity derived from MI analysis, using a regularized model where the 6 CSP components are calculated for each subject. The patterns reveal variations in cortical activation among subjects, highlighting lateralization and individual differences in the motor areas involved during MI tasks.

**Figure 10 sensors-25-02259-f010:**
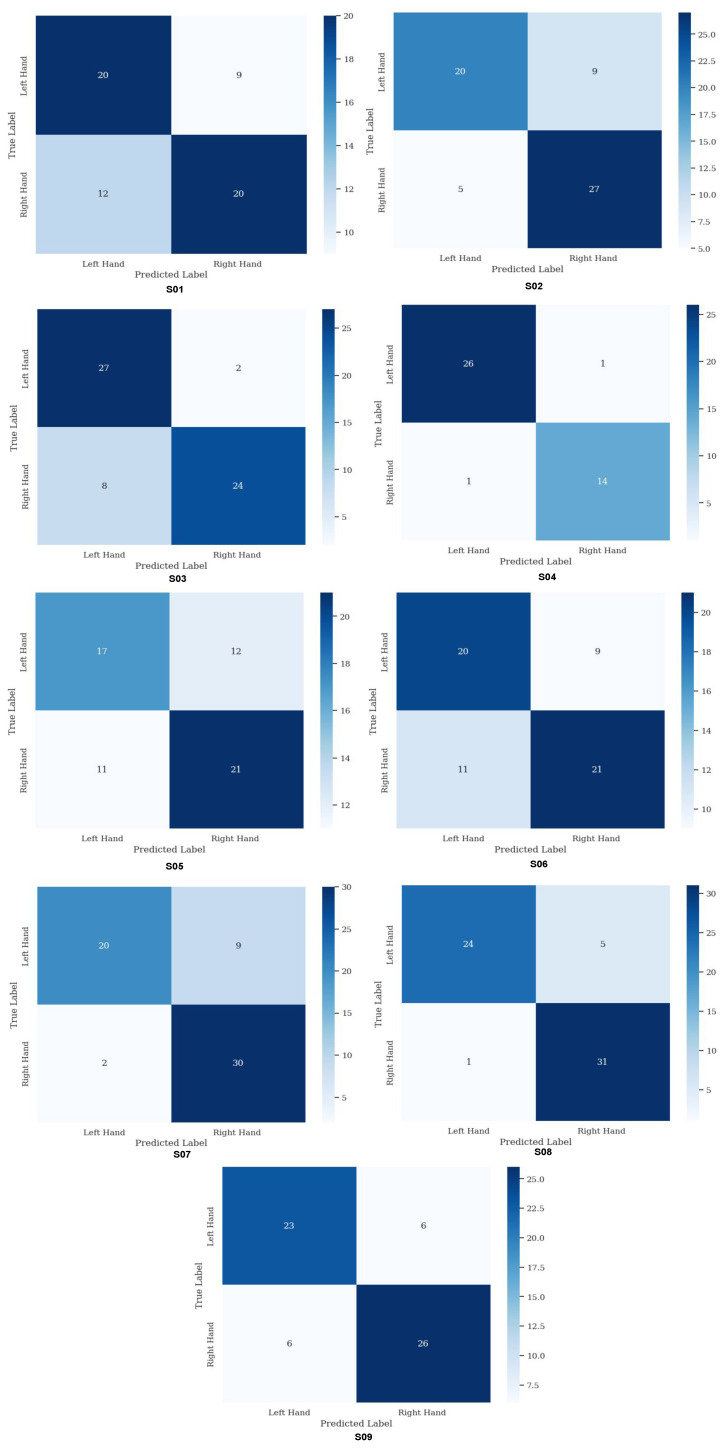
Average confusion matrices across all subjects.

**Table 1 sensors-25-02259-t001:** Table of determination coefficients by subject—elastic net.

Subjects	Coefficient of Determination (r^2^)
Subject 1	0.576
Subject 2	0.641
Subject 3	0.590
Subject 4	0.568
Subject 5	0.613
Subject 6	0.567
Subject 7	0.567
Subject 8	0.595
Subject 9	0.568
**Average**	**0.587**

**Table 2 sensors-25-02259-t002:** Description of the results of the CSP components of the subjects.

Subject	CSP0	CSP1	CSP2	CSP3	CSP4	CSP5
Subject 1	Bilateral distribution	Moderate activation	Uniform distribution	Moderate activity	Lateral activation	Low activation
Subject 2	Strong frontal and lateral activation	Weak central activation	Low distribution	Low activation	Low activation	Low activation
Subject 3	Moderate lateral activation	Diffuse activation	Lateralized activation	Strong activation	Strong lateral activation	Parietal high activity
Subject 4	Uniform distribution	Bilateral activation	Moderate activation	Low activation	Bilateral activation	Bilateral activation
Subject 5	Strong parietal activation	Moderate activation	Diffuse activation	Strong lateral activation	High activity	High activity
Subject 6	Diffuse activation	Moderate lateral activation	Low activation	Diffuse distribution	Low activation	Low activation
Subject 7	Strong negative lateral activation	Motor areas activation	Moderate lateralization	Diffuse activation	Bilateral activation	Strong lateral activation
Subject 8	Centralized activation in CSP1 and CSP3	Diffuse activation	Diffuse activation	Primary motor areas activation	Moderate activation	Lateral areas activation
Subject 9	Symmetry in activations	Moderate activation	Moderate lateral activation	Moderate bilateral activation	Symmetry in activations	Symmetry in activations

**Table 3 sensors-25-02259-t003:** Percentage of correct answers of the SVM classification model for each subject.

Subjects	Accuracy %	Kappa
Subject 1	65.57	0.313
Subject 2	77.05	0.536
Subject 3	83.61	0.674
Subject 4	95.24	0.891
Subject 5	62.30	0.242
Subject 6	67.21	0.344
Subject 7	81.97	0.634
Subject 8	90.16	0.801
Subject 9	80.33	0.605
**Average**	**78.16**	**0.56**

**Table 4 sensors-25-02259-t004:** Performance of the classification model with respect to the MI tasks of each subject.

Metrics	S01	S02	S03	S04	S05	S06	S07	S08	S09	Class	Average
Precision	0.62	0.80	0.77	0.96	0.61	0.65	0.91	0.96	0.79	1	**0.785**
	0.69	0.75	0.92	0.93	0.64	0.70	0.77	0.86	0.81	2	**0.785**
Recall	0.69	0.69	0.93	0.96	0.59	0.69	0.69	0.83	0.79	1	**0.762**
	0.62	0.84	0.75	0.93	0.66	0.66	0.94	0.97	0.81	2	**0.797**
F1-score	0.66	0.74	0.84	0.96	0.60	0.67	0.78	0.89	0.79	1	**0.770**
	0.66	0.79	0.83	0.93	0.65	0.68	0.95	0.91	0.81	2	**0.801**
Support	29	29	29	27	29	29	29	29	29	1	**28.77**
	32	32	32	15	32	32	32	32	32	2	**30.11**

**Table 5 sensors-25-02259-t005:** Comparative analysis of EEG classification methods using BCI Competition IV Dataset IIa for left and right hand motor imagery.

Study	Number of Channels	Methodology	Accuracy (%)
**Our Study**	**8**	**Elastic Net Regression**	**78.16**
Tangermann et al. [65]	22	CSP + LDA	70.1
Ayoobi and Sadeghian [66]	22	Self-Attention Mechanism	74.3
Korkan et al. [67]	22	Deep Neural Network	81.8

**Table 6 sensors-25-02259-t006:** Comparison of EEG-MI classification with CSP/SVM and deep learning methods.

Method	Advantages	Disadvantages	Hyperparameters
CSP + SVM	Low computational cost, interpretable, effective with small datasets [68].	Limited for non-stationary data, does not capture complex spatial relationships.	Regularization parameter (C), Kernel type, Gamma parameter
CNN	Automatically learns spatial features, robust to noise [69].	Requires large datasets and higher computational power.	Learning rate, Batch size, Kernel size, Number of convolutional layers, Optimizer type
RNN (LSTM, GRU)	Captures temporal dependencies in EEG signals [70].	Computationally expensive and may overfit with small datasets.	Number of hidden units, Learning rate, Dropout rate, Sequence length, Optimizer type
Transformer	Models long-range relationships in EEG signals [71].	High computational demand, low interpretability.	Number of attention heads, Number of hidden layers, Learning rate, Batch size, Positional encoding type

## Data Availability

The databases used in this study are public and can be found at the following links: BCI Competition IV Dataset IIa: https://www.bbci.de/competition/iv/ [62], accessed on 1 February 2025.

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
