# Peer review of "EEG Signal Prediction for Motor Imagery Classification in Brain–Computer Interfaces"

_sensors, 2025, doi:10.3390/s25072259_

Round 1
Reviewer 1 Report
Comments and Suggestions for Authors
Introduction and Background
The introduction outlines the challenges faced by EEG-based BCIs, but key references are missing (e.g., citations marked as [?]). Specific missing references include:
- Support for claims regarding motor imagery mechanisms (e.g., Jeannerod, 2001 or similar literature).
- Previous studies on the BCI illiteracy rate (15-30%) and Elastic Net/CSP in EEG.
Methodology Clarity
Elastic Net Implementation:
- Parameters (e.g., alpha=1, l1_ratio=0.5) are stated but the rationale for these choices is not explained. Clarify how hyperparameters are tuned (e.g., cross-validation).
Time Delay τ:
- The manuscript does not explain how τ is chosen. Please provide the rationale or empirical validation for this parameter.
CSP and SVM:
- While CSP/SVM is a standard method, including a comparison with other feature extraction/classification methods (e.g., deep learning) would help contextualize the performance.
Results Presentation
Figures 7-9:
- Ensure the final submission of the images has high resolution. The current description (e.g., "FCz, FC2, FC1") is not clear enough without visual aids.
Subject Variability:
- Address the wide accuracy range (62.30%-95.24%) and discuss potential reasons (e.g., neurophysiological differences between subjects, data quality).
Discussion and Limitations
- Compare the results with recent studies using deep learning (e.g., CNNs for EEG prediction). Highlight the pros and cons of Elastic Net in this context.
- Acknowledge limitations (e.g., dependency on central channels, generalizability to other EEG datasets).
Ethical and Formatting Issues
References [?]: Replace placeholder citations (e.g., [?]) with full references.Language: Modify the grammar for clarity (e.g., "This requirement posed significant obstacles...").Data Availability: Confirm that the BCI Competition IV dataset IIa is publicly accessible and cite its primary source (e.g., Brunner et al., 2008).
Reviewer 2 Report
Comments and Suggestions for Authors
The study “EEG Signal Prediction for Motor Imagery Classification in Brain-Computer Interfaces” presents a novel signal prediction method aimed at achieving high accuracy in classifying motor imagery based on EEG signals collected from a limited number of electrodes. The authors developed a signal prediction model using the Elastic Net regression approach. This model estimates EEG signals from 22 full channels using only eight centrally placed channels. The results indicate that this prediction method is significantly effective, demonstrating an average classification accuracy of 78.16%. This technique seems to deliver precise estimates with only a limited number of electrodes, and it has potential for real-world applications in motor imagery-based brain-computer interfaces, addressing time and cost challenges often associated with systems requiring high electrode density. Moreover, the manuscript is well-written and organized, and I highly recommend its publication.
Minor points:
Can you include the effect of each EEG sensor’s impedance in the introduction? This electrical impedance should impact the quality of the signal. The impedance depends on the sensor’s connection to the subject’s scalp, among other variables.
In Equation (1), the parameter Tau should be defined as an integer. Otherwise, you could not use it in your algorithm.
In Figure 10. Please increase the size of the axis titles; they are hard to read.
You claim that the proposed method outperforms traditional approaches that utilize fewer electrodes, offering improved outcomes compared to methods that rely on full-channel EEG. Can you please add a comparative table with references proving this assertion?
Round 2
Reviewer 1 Report
Comments and Suggestions for Authors
Thank you for the author's response to my concern, I believe it can be published.